# LARGE-SCALE VISUAL SPEECH RECOGNITION

## ABSTRACT

This work presents a scalable solution to continuous visual speech recognition. To achieve this, we constructed the largest existing visual speech recognition dataset, consisting of pairs of text and video clips of faces speaking (3,886 hours of video). In tandem, we designed and trained an integrated lipreading system, consisting of a video processing pipeline that maps raw video to stable videos of lips and sequences of phonemes, a scalable deep neural network that maps the lip videos to sequences of phoneme distributions, and a production-level speech decoder that outputs sequences of words. The proposed system achieves a word error rate (WER) of $40.9\%$ as measured on a held-out set. In comparison, professional lipreaders achieve either $86.4\%$ or $92.9\%$ WER on the same dataset when having access to additional types of contextual information. Our approach significantly improves on previous lipreading approaches, including variants of *LipNet* and of *Watch, Attend, and Spell* (WAS), which are only capable of $89.8\%$ and $76.8\%$ WER respectively.

## 1 INTRODUCTION AND MOTIVATION

Deep learning techniques have allowed for significant advances in lipreading over the last few years (Assael et al., 2017; Chung et al., 2017; Thanda & Venkatesan, 2017; Koumparoulis et al., 2017; Chung & Zisserman, 2017; Xu et al., 2018). However, these approaches have often been limited to narrow vocabularies, and relatively small datasets (Assael et al., 2017; Thanda & Venkatesan, 2017; Xu et al., 2018). Often the approaches focus on single-word classification (Hinton et al., 2012; Chung & Zisserman, 2016a; Wand et al., 2016; Stafylakis & Tzimiropoulos, 2017; Ngiam et al., 2011; Sui et al., 2015; Ninomiya et al., 2015; Petridis & Pantic, 2016; Petridis et al., 2017; Noda et al., 2014; Koller et al., 2015; Almajai et al., 2016; Takashima et al., 2016; Wand & Schmidhuber, 2017) and do not attack the continuous recognition setting. In this paper, we contribute a novel method for large-vocabulary continuous visual speech recognition. We report substantial reductions in word error rate (WER) over the state-of-the-art approaches even with a larger vocabulary.

Assisting people with speech impairments is a key motivating factor behind this work. Visual speech recognition could positively impact the lives of hundreds of thousands of patients with speech impairments worldwide. For example, in the U.S. alone 103,925 tracheostomies were performed in 2014 (HCUPnet, 2014), a procedure that can result in a difficulty to speak (disphonia) or an inability to produce voiced sound (aphonia). While this paper focuses on a scalable solution to lipreading using a vast diverse dataset, we also expand on this important medical application in Appendix A. The discussion there has been provided by medical experts and is aimed at medical practitioners.

We propose a novel lipreading system, illustrated in Figure 1, which transforms raw video into a word sequence. The first component of this system is a data processing pipeline used to create the *Large-Scale Visual Speech Recognition* (LSVSR) dataset used in this work, distilled from YouTube videos and consisting of phoneme sequences paired with video clips of faces speaking (3,886 hours of video). The creation of the dataset alone required a non-trivial combination of computer vision and machine learning techniques. At a high-level this process takes as input raw video and annotated audio segments, filters and preprocesses them, and produces a collection of aligned phoneme and lip frame sequences. Compared to previous work on visual speech recognition, our pipeline uses landmark smoothing, a blurriness filter, an improved speaking classifier network and outputs phonemes. The details of this process are described in Section 3.

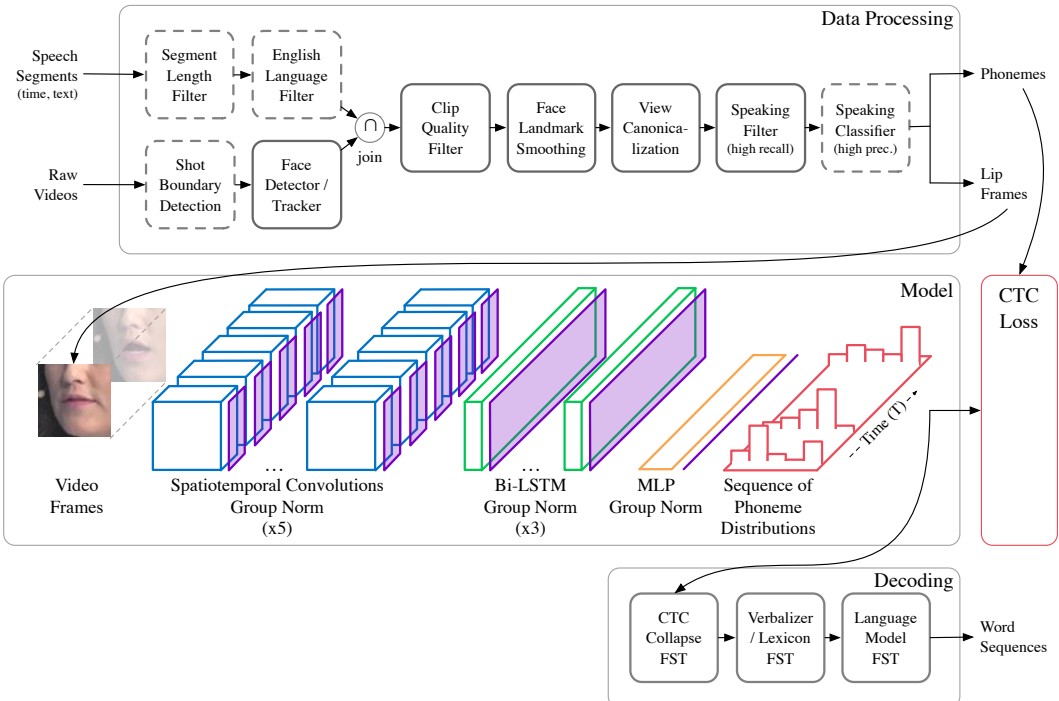

Figure 1: The full visual speech recognition system introduced by this work consists of a data processing pipeline that generates lip and phoneme clips from YouTube videos (see Section 3), and a scalable deep neural network for phoneme recognition combined with a production-grade word-level decoding module used for inference (see Section 4).

Next, this work introduces a new neural network architecture for lipreading, which we call *Vision to Phoneme* (V2P), trained to produce a sequence of phoneme distributions given a sequence of video frames. In light of the large scale of our dataset, the network design has been highly tuned to maximize predictive performance subject to the strong computational and memory limits of modern GPUs in a distributed setting. In this setting we found that techniques such as group normalization (Wu & He, 2018) to be key to the reported results. Furthermore, our approach is the first to combine a deep learning-based visual speech recognition model with production-grade word-level decoding techniques. By decoupling phoneme prediction and word decoding as is often done in speech recognition, we are able to arbitrarily extend the vocabulary without retraining the neural network. Details of our model and this decoding process are given in Section 4. By design, the trained model only performs well under optimal lighting conditions, within a certain distance from a subject, and at high quality. It does not perform well in other contexts.

Finally, this entire lipreading system results in an unprecedented WER of 40.9% as measured on a held-out set from our dataset. In comparison, professional lipreaders achieve either 86.4% or 92.9% WER on the same dataset, depending on the amount of context given. Similarly, previous state-of-the-art approaches such as variants of *LipNet* Assael et al. (2017) and of *Watch, Attend, and Spell* (WAS) Chung et al. (2017) demonstrated WERs of only 89.8% and 76.8% respectively.

## 2 RELATED WORK

While there is a large body of literature on automated lipreading, much of the early work focused on single-word classification and relied on substantial prior knowledge (Chu & Huang, 2000; Matthews et al., 2002; Pitsikalis et al., 2006; Lucey & Sridharan, 2006; Papandreou et al., 2007; Zhao et al., 2009; Gurban & Thiran, 2009; Papandreou et al., 2009). For example, Goldschen et al. (1997) predicted continuous sequences of tri-visemes using a traditional HMM model with visual features extracted from a codebook of clustered mouth region images. The predicted visemes were used to distinguish sentences from a set of 150 possible sentences. Furthermore, Potamianos et al. (1997)

predict words and sequences digits using HMMs, Potamianos & Graf (1998) introduce multi-stream HMMs, and Potamianos et al. (1998) improve the performance by using visual features in addition to the lip contours. Later, Chu & Huang (2000) used coupled HMMs to jointly model audio and visual streams to predict sequences of digits. Neti et al. (2000) used HMMs for sentence-level speech recognition in noisy environments of the IBM ViaVoice dataset by fusing handcrafted visual and audio features. More recent attempts using traditional speech, vision and machine learning pipelines include the works of Gergen et al. (2016); Paleček (2017); Hassanat (2011) and Bear & Harvey (2016). For further details, we refer the reader to the survey material of Potamianos et al. (2004) and Zhou et al. (2014).

However, as noted by Zhou et al. (2014) and Assael et al. (2017), until recently generalization across speakers and extraction of motion features have been considered open problems. Advances in deep learning have made it possible to overcome these limitations, but most works still focus on single-word classification, either by learning visual-only representations (Hinton et al., 2012; Chung & Zisserman, 2016a; Wand et al., 2016; Stafylakis & Tzimiropoulos, 2017; Wand & Schmidhuber, 2017), multimodal audio-visual representations (Ngiam et al., 2011; Sui et al., 2015; Ninomiya et al., 2015; Petridis & Pantic, 2016; Petridis et al., 2017), or combining deep networks with traditional speech techniques (e.g. HMMs and GMM-HMMs) (Noda et al., 2014; Koller et al., 2015; Almajai et al., 2016; Takashima et al., 2016).

LipNet (Assael et al., 2017) was the first end-to-end model to tackle sentence-level lipreading by predicting character sequences. The model combined spatiotemporal convolutions with gated recurrent units (GRUs) and was trained using the CTC loss function. LipNet was evaluated on the GRID corpus (Cooke et al., 2006), a limited grammar and vocabulary dataset consisting of 28 hours of 5-word sentences, where it achieved $4.8\%$ and $11.4\%$ WER in overlapping and unseen speaker evaluations respectively. By comparison, the performance of competent human lipreaders on GRID was $47.7\%$. LipNet is the closest model to our neural network. Several similar architectures were subsequently introduced in the works of Thanda & Venkatesan (2017) who study audio-visual feature fusion, Koumparoulis et al. (2017) who work on a small subset of 18 phonemes and 11 words to predict digit sequences, and Xu et al. (2018) who presented a model cascading CTC with attention.

Chung et al. (2017) were the first to use sequence-to-sequence models with attention to tackle audio-visual speech recognition with a real-world dataset. The model "Watch, Listen, Attend and Spell" (WLAS), consists of a visual (WAS) and an audio (LAS) module. To evaluate WLAS, the authors created LRS, the largest dataset at that point with approximately 246 hours of clips from BBC news broadcasts, and introduced an efficient video processing pipeline. The authors reported $50.2\%$ WER, with the performance of professional lipreaders being $87.6\%$ WER. Chung & Zisserman (2017) extended the work to multi-view sentence-level lipreading, achieving $62.8\%$ WER for profile views and $56.4\%$ WER for frontal views. Both Chung et al. (2017) and Chung & Zisserman (2017) pre-learn features with the audio-video synchronization classifier of Chung & Zisserman (2016b), and fix these features in order to compensate for the large memory requirements of their attention networks. Contemporaneously with our work, Afouras et al. (2018c) presented LRS3-TED, a dataset generated from English language talks available online. Using pre-learned features Afouras et al. (2018a) presented a seq2seq and a CTC architecture based on character-level self-attention transformer models. On LRS3-TED, these models achieved a WER of $57.9\%$ and $61.8\%$ respectively. Other related advances include works using vision for silent speech reconstruction (Le Cornu & Milner, 2017; Ephrat & Peleg, 2017; Akbari et al., 2017; Gabbay et al., 2017) and for separating an audio signal to individual speech sources (Ephrat et al., 2018; Afouras et al., 2018b).

In contrast to the approach of Assael et al. (2017), our model (V2P) uses a network to predict a sequence of phoneme distributions which are then fed into a decoder to produce a sequence of words. This flexible design enables us to easily accommodate very large vocabularies, and in fact we can extend the size of the vocabulary without having to retrain the deep network. Unlike previous work, V2P is memory and computationally efficient without requiring pre-trained features (Chung et al., 2017; Chung & Zisserman, 2017).

Finally, the data processing pipeline used in this work results in a significantly larger and more diverse training dataset than in all previous efforts. While the first large-vocabulary lipreading dataset was IBM ViaVoice (Neti et al., 2000), more recently the far larger LRS and MV-LRS datasets (Chung et al., 2017; Chung & Zisserman, 2017) were generated from BBC news broadcasts, and the LRS3-TED dataset was generated from conference talks. MV-LRS and LRS3-TED are the only publicly

| Dataset | Utter. | Hours | Vocab |
|---|---|---|---|
| GRID | 33,000 | 28 | 51 |
| IBM ViaVoice | 17,111 | 35 | 10,400 |
| MV-LRS | 74,564 | ∼155 | 14,960 |
| LRS | 118,116 | ∼246 | 17,428 |
| LRS3-TED | ∼165,000 | ∼475 | ∼57,000 |
| **LSVSR** | **2,934,899** | **3,886** | **127,055** |

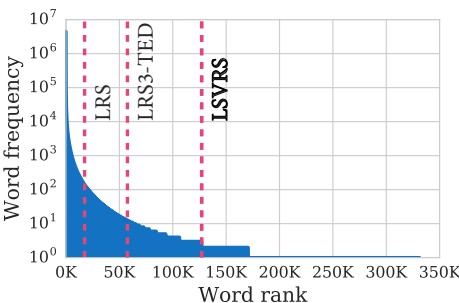

Figure 2: Left: A comparison of sentence-level (word sequence) visual speech recognition datasets. Right: Frequency of words in the LSVSR dataset in decreasing order of occurrence; approximately 350K words occur at least 3 times. We used this histogram to select a vocabulary of 127,055 words as it captures most of the mass.

available large-vocabulary datasets, although both are limited to academic usage. In comparison, our dataset (LSVSR) is an order of magnitude greater than any previous dataset with 3,886 hours of audio-video-text pairs. In addition, the content is much more varied (i.e. not news-specific), resulting in a 2.2× larger vocabulary of 127,055 words. Figure 2 shows a comparison of sentence-level (word sequence) visual speech recognition datasets.

## 3 A DATA PIPELINE FOR LARGE-SCALE VISUAL SPEECH RECOGNITION

In this section we discuss the data processing pipeline, again illustrated in Figure 1, used to create the LSVSR dataset. Our pipeline makes heavy use of large-scale parallel processing and is implemented as a number of independent modules and filters on top of FlumeJava (Chambers et al., 2010). In particular, our dataset is extracted from public YouTube videos. This is a common strategy for building datasets in ASR and speech enhancement (Liao et al., 2013; 2015; Kuznetsov et al., 2016; Soltau et al., 2017; Ephrat et al., 2018).

In our case, we build on the work of Liao et al. (2013) to extract audio clips paired with transcripts, yielding 140,000 hours of audio segments. After post-processing we obtain a dataset consisting of paired video and phoneme sequences, where video sequences are represented as identically-sized frames (here, $128 \times 128$) stacked in the time-dimension. Although our pipeline is used to process clips pre-selected from YouTube (Liao et al., 2013), only about 2% of clips satisfy our filtering criteria. Finally, by eliminating the components marked by dashes in Figure 1, i.e. those components whose primary use are in producing paired training data, this same pipeline can be used in combination with a trained model to predict word sequences from raw videos. In what follows we describe the individual components that make up this pipeline.

**Length filter, language filter.** The duration of each segment extracted from YouTube is limited to between 1 and 12 seconds, and the transcripts are filtered through a language classifier (Salcianu et al., 2018) to remove non-English utterances. For evaluation, we further remove the utterances containing fewer than 6 words. Finally, the aligned phoneme sequences are obtained via a standard forced alignment approach using a lexicon with multiple pronunciations (Liao et al., 2013). The phonetic alphabet is a reduced version of X-SAMPA (Wells, 1995) with 40 phonemes plus silence.

**Raw videos, shot boundary detection, face detection.** Constant spatial padding in each video segment is eliminated before a standard, thresholding color histogram classifier (Mas & Fernandez, 2003) identifies and removes segments containing shot boundaries. FaceNet (Schroff et al., 2015) is used to detect and track faces in every remaining segment.

**Clip quality filter.** Speech segments are joined with the set of tracked faces identified in the previous step and filtered based on the quality of the video, removing blurry clips and clips including faces with an eye-to-eye width of less than 80 pixels. Frame rates lower than 23fps are also eliminated (Saitoh & Konishi, 2010; Taylor et al., 2014). We allow a range of input frame rates—varying frame rates has an effect similar to different speaking paces—however, frame rates above 30fps are downsampled.

**Face landmark smoothing.** The segments are processed by a face landmark tracker and the resulting landmark positions are smoothed using a temporal Gaussian kernel. Intuitively, this simplifies learning filters for the 3D convolution layers by reducing spatiotemporal noise. Empirically, our preliminary studies showed smoothing was crucial for achieving optimal performance. Next, following previous literature (Chung et al., 2017), we keep segments where the face yaw and pitch remain within $\pm 30°$. Models trained outside this range perform worse (Chung & Zisserman, 2017).

**View canonicalization.** We obtain canonical faces using a reference canonical face model and by applying an affine transformation on the landmarks. Then, we use a thumbnail extractor which is configured to crop the area around the lips of the canonical face.

**Speaking filter.** Using the extracted and smoothed landmarks, minor lip movements and non-speaking faces are discarded using a threshold filter. This process involves computing the mouth openness in all frames, normalizing by the size of the face bounding box, and then thresholding on the standard deviation of the normalized openness. This classifier has very low computational cost, but high recall, e.g. voice-overs are not handled. It's contribution was very important to process approximately 16 years of audio-video-text pairs within reasonable time.

**Speaking classifier.** As a final step, we build *V2P-Sync*, a neural network architecture to verify the audio and video channel alignment inspired by the work of Chung & Zisserman (2016b) and Torfi et al. (2017). However, V2P-Sync takes advantage of face landmark smoothing, 3D convolutions, and high resolution inputs. V2P-Sync uses longer time segments as inputs and spatiotemporal convolutions as compared to the spatial-only convolutions of Chung & Zisserman, and view canonicalization and higher resolution inputs ($128 \times 128$ vs $100 \times 60$) as compared to Torfi et al.. These characteristics facilitate the extraction of temporal features which is key to our task. V2P-Sync, takes as input a pair of a $\log$ mel-spectrogram and 9 grayscale video frames and produces an embedding for each using two separate neural network architectures. If the Euclidean distance of the audio and video embeddings is less than a given threshold the pair is classified as synchronized. The architecture is trained using a contrastive loss similar to Chung & Zisserman. Since there is no labeled data for training, the initial unfiltered pairs are used as positive samples with negative samples generated by randomly shifting the video of an unfiltered pair. After convergence the dataset is filtered using the trained model, which is then fine-tuned on the resulting subset of the initial dataset. The final model is used to filter the dataset a second time, achieving an accuracy of $81.2\%$. This accuracy is improved as our audio-video pairs are processed by sliding V2P-Sync on 100 equally spaced segments and their scores are averaged. For further architectural details, we refer the reader to Appendix F.

## 4 AN EFFICIENT SPATIOTEMPORAL MODEL OF VISUAL SPEECH RECOGNITION

This work introduces the V2P model, which consists first of a *3d convolutional module* for extracting spatiotemporal features from a given video clip. These features are then aggregated over time with a *temporal module* which outputs a sequence of phoneme distributions. Given input video clips and target phoneme sequences the model is trained using the *CTC* loss function. Finally, at test-time, a *decoder* based on finite state transducers (FSTs) is used to produce a word sequence given a sequence of phoneme distributions. For further details we refer the reader to Appendix G.

**Neural network architecture.** Although the use of optical-flow filters as inputs is commonplace in lipreading (Mase & Pentland, 1991; Gray et al., 1997; Yoshinaga et al., 2003; Tamura et al., 2004; Wang et al., 2008; Shaikh et al., 2010), in this work we designed a vision module based on VGG (Simonyan & Zisserman, 2015) to explicitly address motion feature extraction. We adapted VGG to make it volumetric, which proved crucial in our preliminary empirical evaluation and has been established in previous literature (Assael et al., 2017). The intuition behind this is the importance of spatiotemporal relationships in human visual speech recognition, e.g. measuring how lip shape changes over time. Furthermore, the receptive field of the vision module is 11 video frames, roughly 0.36–0.44 seconds, or around twice the typical duration of a phoneme.

One of the main challenges in training a large vision module is finding an effective balance between performance and the imposed constraints of GPU memory. Our vision module consists of 5 convolutional layers with $[64, 128, 256, 512, 512]$ filters. By profiling a number of alternative architectures, we found that high memory usage typically came from the first two convolutional layers. To reduce the memory footprint we limit the number of convolutional filters in these layers, and since the frame

is centered around the lips, we omit spatial padding. Since phoneme sequences can be quite long, but with relatively low frame rate (approximately 25–30 fps), we maintain padding in the temporal dimension and always convolve with unit stride in order to avoid limiting the number of output tokens. Despite tuning the model to reduce the number of activations, we are still only able to fit 2 batch elements on a GPU. Hence, we distribute training across 64 workers in order to achieve a batch size of 128. Due to communication costs, batch normalization is expensive if one wants to aggregate the statistics across all workers, and using only two examples per batch results in noisy normalization statistics. Thus, instead of batch normalization, we use group normalization (Wu & He, 2018), which divides the channels into groups and computes the statistics within these groups. This provides more stable learning regardless of batch size.

The outputs of the convolutional stack are then fed into a temporal module which performs longer-scale aggregation of the extracted features over time. In constructing this component we evaluated a number of recurrent neural network and dilated convolutional architectures, the latter of which are evaluated later as baselines. The best architecture presented performs temporal aggregation using a stack of 3 bidirectional LSTMs (Hochreiter & Schmidhuber, 1997) with a hidden state of 768, interleaved with group normalization. The output of these LSTM layers is then fed through a final MLP layer to produce a sequence of exactly $T$ conditionally independent phoneme distributions $p(u_t|\mathbf{x})$. This entire model is then trained using the CTC loss we describe next.

This model architecture is similar to that of the closest related work, LipNet (Assael et al., 2017), but differs in a number of crucial ways. In comparison to our work, LipNet used GRU units and dropout, both of which we found to perform poorly in preliminary experiments. Our model is also much bigger: LipNet consists of only 3 convolutional layers of $[32, 64, 96]$ filters and 3 GRU layers with hidden state of size 256. Although the small size of LipNet means that it does not require any distributed computation to reach effective batch sizes, we will see that this drop in size coincides with a similar drop in performance. Finally, while both models use a CTC loss for training, the architecture used in V2P is trained to predict phonemes rather than characters; as we argue shortly this provides V2P with a much simpler mechanism for representing word uncertainty.

**Connectionist temporal classification (CTC).** CTC is a loss function for the parameterization of distributions over sequences of label tokens, without requiring alignments of the input sequence to the label tokens (Graves et al., 2006). To see how CTC works, let $V$ denote the set of single-timestep label tokens. To align a label sequence with size-$T$ sequences given by the temporal module, CTC allows the model to output blank symbols ␣ and repeat consecutive symbols. Let the function $\mathcal{B} : (V \cup \{␣\})^* \rightarrow V^*$ be defined such that, given a string potentially containing blank tokens, it deletes adjacent duplicate characters and removes any blanks. The probability of observing label sequence $y$ can then be obtained by marginalizing over all possible alignments of this label, $p(y|\mathbf{x}) = \sum_{u \in \mathcal{B}^{-1}(y)} p(u_1|\mathbf{x}) \cdots p(u_T|\mathbf{x})$, where $\mathbf{x}$ is input video. For example, if $T = 5$ the probability of sequence 'bee' is given by $p(be␣e␣) + p(␣be␣e) + \cdots + p(bbe␣e) + p(be␣ee)$. Note that there must be a blank between the 'e' characters to avoid collapsing the sequence to 'be'.

Since CTC prevents us from using autoregressive connections to handle inter-timestep dependencies of the label sequence, the marginal distributions produced at each timestep of the temporal module are conditionally independent, as pointed out above. Therefore, to restore temporal dependency of the labels at test-time, CTC models are typically decoded with a beam search procedure that combines the probabilities with that of a language model.

**Rationale for phonemes and CTC.** In speech recognition, whether on audio or visual signals, there are two main sources of uncertainty: uncertainty in the sounds that are in the input, and uncertainty in the words that correspond to these sounds. This suggests modelling

$$p(\text{words}|\mathbf{x}) = \sum_{\text{phonemes}} p(\text{words}|\text{phonemes})p(\text{phonemes}|\mathbf{x}) \approx p(\text{words}|\text{phonemes})p(\text{phonemes}|\mathbf{x}),$$

where the approximation is by the assumption that a given word sequence often has a single or dominant pronunciation. While previous work uses CTC to model characters given audio or visual input directly (Assael et al., 2017; Amodei et al., 2016), we argue this is problematic as the conditional independence of CTC timesteps means that the temporal module must assign a high probability to a single sequence in order to not produce spurious modes in the CTC distribution.

To explain why modeling characters with CTC is problematic, consider two character sequences "fare" and "fair" that are homophones, i.e. they have the same pronunciation (i.e. /fɛː/). The difficulty we will describe is independent of the model used, so we will consider a simple unconditional model where each character $c$ is assigned probability given by the parameters $\pi_t^c = P(u_t = c)$ and the probability of a sequence is given by its product, e.g. $p(\text{fare}) = \pi_1^f \pi_2^a \pi_3^r \pi_4^e$. The maximum likelihood estimate, $\arg\max_\pi p(\text{fare})p(\text{fair})$, however, assigns equal 1/4 probability to each of "fare", "fair", "faie", "farr", as shown in Figure 3, resulting in two undesirable words. Ultimately this difficulty arises due to the independence assumption of CTC

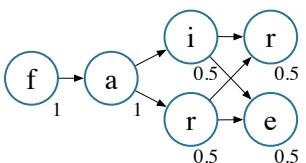

Figure 3: Example homophone issue when modelling characters with CTC.

and the many-to-many mapping of characters to words[1]. This same difficulty arises if we replace the parameters above with the outputs of a network mapping from videos to tokens.

An additional source of uncertainty in visual speech recognition is introduced by the fact that the information required to disambiguate different phonemes is not visible (e.g. the position of the tongue). The resulting visually similar phonemes are called visemes. This can be seen in Appendix C where the ratio of insertions and substitutions when computing the edit distance of a visual model is substantially higher than the ratio of an audio model trained on the same dataset. Furthermore, recent literature suggests that phoneme-to-viseme mappings may differ per speaker (Bear & Harvey, 2017), making it difficult to incorporate this knowledge in the design. Thus, using phonemes, which have a one-to-many mapping to words, allows the temporal model to only model visual uncertainty, and the word uncertainty can instead be handled by the decoder described below.

Alternatively to using phonemes with CTC, some previous work solves this problem using RNN transducers (Rao et al., 2017) or sequence-to-sequence with attention (Chung et al., 2017), which jointly model all sources of uncertainty. However, Prabhavalkar et al. (2017) showed in the context of acoustic speech recognition that these models were unable to significantly outperform a baseline CTC model (albeit using context-dependent phonemes and further sequence-discriminative training) when combined with a decoding pipeline similar to ours. Hence, for reasons of performance and easier model training, especially important with our large model, we choose to output phonemes rather than words or characters directly. Additionally, and crucial for many applications, CTC also provides extra flexibility over alternatives. The fact that the lexicon (phoneme to word mapping) and language model are separate and part of the decoder, affords one the ability to trivially change the vocabulary and language model (LM) arbitrarily. This allows for visual speech recognition in narrower domains or updating the vocabulary and LM with new words without requiring retraining of the phoneme recognition model. This is nontrivial in other models, where the language model is part of the RNN.

**Decoding.** As described earlier, our model produces a sequence of phoneme distributions; given these distributions we use an industry-standard decoding method using finite state transducers (FSTs) to arrive at word sequences. Such techniques are extensively used in speech recognition (e.g. Miao et al., 2015; McGraw et al., 2016); we refer the reader to the thorough presentation of Mohri et al. (2002). In our work we make use of a combination of three individual (weighted) FSTs, or WFSTs. The first *CTC postprocessing FST* removes duplicate symbols and CTC blanks. Next, a *lexicon* FST maps input phonemes to output words. Third, an *n-gram language model* with backoff can be represented as a WFST from words to words. In our case, we use a 5-gram model with Katz backoff with about 50 million n-grams and a vocabulary size of about one million. The composition of these three FSTs results another WFST transducing from phoneme sequences to (reweighted) word sequences. Finally, a search procedure is employed to find likely word sequences from phoneme distributions.

## 5 EVALUATION

We examine the performance of V2P trained on LSVSR with hyperparameters tuned on a validation set. We evaluate it on a held-out test set roughly 37 minutes long, containing approximately 63,000 video frames and 7100 words. We also describe and compare against a number of alternate methods from previous work. In particular, we show that our system gives significant performance improvements over professional lipreaders as well previous state-of-the-art methods for visual speech

---

[1]Languages such as Korean, where there is a one-to-one correspondence between pronunciation and orthography, do not give rise to such discrepancies.

recognition. Except for V2P-NoLM, all models used the same 5-gram word-level language model during decoding. To construct the validation and test sets we removed blurry videos by thresholding the variance of the Laplacian of each frame (Pech-Pacheco et al., 2000); we kept them in the training set as a form of data augmentation.

**Professional lipreaders.** We consulted a professional lipreading company to measure the difficulty of LSVSR and hence the impact that such a model could have. Since the inherent ambiguity in lipreading necessitates relying on context, we conducted experiments both with and without context. In both cases we generate modified clips from our test set, but cropping the whole head in the video, as opposed to just the mouth region used by our model. The lipreaders could view the video up to 10 times, at half or normal speed each time. To measure without-context performance, we selected clips with transcripts that had at least 6 words. To measure how much context helps performance, we selected clips with at least 12 words, and presented to the lipreader the first 6 words, the title, and the category of the video, then asked them to transcribe the rest of the clip. The lipreaders transcribed a subset of our test set containing 153 and 274 videos with and without context, respectively.

**Audio-Ph.** For an approximate bound on performance, we train an audio speech recognition model on the audio of the utterances. The architecture is based on Deep Speech 2 (Amodei et al., 2016), but trained to predict phonemes rather than characters.

**Baseline-LipNet-Ch.** Using our training setup, we replicate the character-level CTC architecture of LipNet (Assael et al., 2017). As with the phoneme models, we use an FST decoding pipeline and the same language model, but instead of a phoneme-based lexicon we use a character-level one as described in Miao et al. (2015).

**Baseline-LipNet-Ph.** We also train LipNet to predict phonemes, still with CTC and using the same FST-based decoding pipeline and language model.

**Baseline-LipNet-Large-Ph.** Recall from the earlier discussion that LipNet uses dropout, whereas V2P makes heavy use of group normalization, crucial for our small batches per worker. For a fair size-wise comparison, we introduce a replica of V2P, that uses GRUs, dropout, and no normalization.

**Baseline-Seq2seq-Ch.** Using our training setup, we compared to a variant of the previous state-of-the-art sequence-to-sequence architecture of WAS that predicts character sequences (Chung et al., 2017). Although their implementation was followed as closely as possible, training end-to-end quickly exceeded the memory limitations of modern GPUs. To work around these problems, the authors kept the convolutional weights fixed using a pretrained network from audio-visual synchronization classification (Chung & Zisserman, 2016b), which we were unable to use as their network inputs were processed differently. Instead, we replace the 2D convolutional network with the *improved* lightweight 3D visual processing network of V2P. From our empirical evaluation, including preliminary experiments not reported here and as shown by earlier work (Assael et al., 2017), we believe that the 3D spatiotemporal aggregation of features benefits performance. After standard beam search decoding, we use the same 5-gram word LM as used for the CTC models to perform reranking.

**V2P-FullyConv.** Identical to V2P, except the LSTMs in the temporal aggregation module are replaced with 6 dilated temporal convolution layers with a kernel size of 3 and dilation rates of [1,1,2,4,8,16], yielding a fully convolutional model with 12 layers.

**V2P-NoLM.** Identical to V2P, except during decoding, where the LM is replaced with a dictionary consisting of 100k words. The words are then weighted by their smoothed frequency in the training data, essentially a uni-gram language model.

## 5.1 RESULTS

Table 1 shows the phoneme error rate, character error rate, and word error rate for all of the models, and the number of parameters for each. The error rates are computed as the sum of the edit distances of the predicted and ground-truth sequence pairs divided by total ground-truth length. We also compute and display the standard error associated with each rate, estimated by bootstrap sampling.

These results show that the variant of LipNet tested in this work is approximately able to perform on-par with professional lipreaders with WER of 86.4% and 89.8% respectively, even when the given professional is given additional context. Similarly, we see that the WAS variant provides a substantial reduction to this error, resulting in a WER of 76.8%. However, the full V2P method presented in this

Table 1: Performance evaluation on LSVSR test set. Columns show phoneme, character, and word error rates, respectively. Standard deviations are bootstrap estimates.

| Method | Params | PER | CER | WER |
|---|---|---|---|---|
| Professional w/o context | − | − | − | $92.9 \pm 0.9$ |
| Professional w/ context | − | − | − | $86.4 \pm 1.4$ |
| Audio-Ph | 58M | $12.5 \pm 0.5$ | $11.5 \pm 0.6$ | $18.3 \pm 0.9$ |
| Baseline-LipNet-Ch | 7M | − | $64.6 \pm 0.5$ | $93.0 \pm 0.6$ |
| Baseline-LipNet-Ph | 7M | $65.8 \pm 0.4$ | $72.8 \pm 0.5$ | $89.8 \pm 0.5$ |
| Baseline-Seq2seq-Ch | 15M | − | $49.9 \pm 0.6$ | $76.8 \pm 0.8$ |
| Baseline-LipNet-Large-Ph | 40M | $53.0 \pm 0.5$ | $54.0 \pm 0.8$ | $72.7 \pm 1.0$ |
| V2P-FullyConv | 29M | $41.3 \pm 0.6$ | $36.7 \pm 0.9$ | $51.6 \pm 1.2$ |
| V2P-NoLM | 49M | $33.6 \pm 0.6$ | $34.6 \pm 0.8$ | $53.6 \pm 1.0$ |
| **V2P** | **49M** | $\mathbf{33.6 \pm 0.6}$ | $\mathbf{28.3 \pm 0.9}$ | $\mathbf{40.9 \pm 1.2}$ |

work is able to further halve the WER, obtaining a value of 40.9% at testing time. Interestingly, we see that although the bi-directional LSTM provides the best performance, using a fully-convolutional network still results in performance that is significantly better than all previous methods. Finally, although we see that the full V2P model performs best, removing the language model results only in a drop of approximately 13 WER to 53.6%.

By predicting phonemes directly, we also side-step the need to design phoneme-to-viseme mappings (Bear & Harvey, 2017). The inherent uncertainty is instead modelled directly in the predictive distribution. For instance, using edit distance alignments of the predictions to the ground-truths, we can determine which phonemes were most frequently erroneously included or missed, as shown in Figure 4. Here we normalize the rates of deletions vs insertions, however empirically we saw that deletions were much more common than inclusions. Among these errors the most common include phonemes that are often occluded by the teeth (/d/, /n/, and /t/) as well as the most common English vowel /@/. Finally, by differentiating the likelihood of the phoneme sequence with respect to the inputs using guided backpropagation (Springenberg et al., 2014), we compute the saliency maps shown in the top row of Figure 5 as a white overlay. The entropy at each timestep of the phoneme predictive distribution is shown as well. A full confusion matrix, absolute deletion/insertion/substitution counts, and additional saliency maps are shown in Appendices B, C and E.

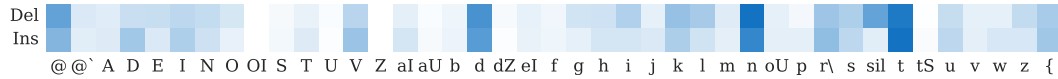

Figure 4: This heatmap shows which insertion and deletion errors were most common on the test set. Blue indicates more insertions or deletions occurred. Substitutions and further details are shown in Appendices B and C.

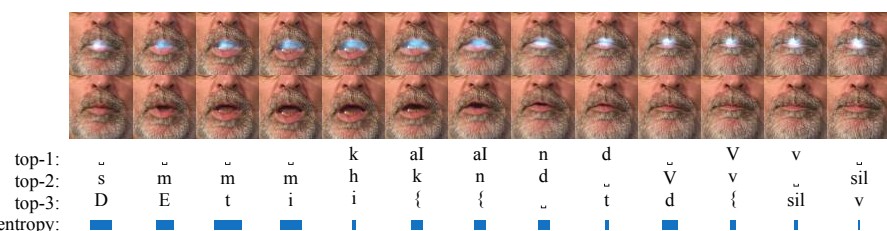

Figure 5: Saliency map for "kind of" and the top-3 predictions of each frame. The CTC blank character is represented by '␣'. The unaligned ground truth phoneme sequence is /k aI n d V v/.

To demonstrate the generalization power of our V2P approach, we also compare it to the results of the TM-seq2seq model of Afouras et al. (2018a) on LRS3-TED (Afouras et al., 2018c). Unlike LSVSR, the LRS3-TED dataset includes faces at angles between $\pm 90°$ instead of $\pm 30°$, and clips may be shorter than one second. Despite the fact that we

Table 2: Evaluation on LRS3-TED.

| Model | Filtered Test | Full Test |
|---|---|---|
| TM-seq2seq | – | 57.9 |
| **V2P** | $\mathbf{47.0 \pm 1.6}$ | $\mathbf{55.1 \pm 0.9}$ |

do not train or fine-tune V2P on LRS3-TED, our approach still outperforms the state-of-the-art model trained on that dataset in terms of test set accuracy. In particular, we conducted two experiments. First, we evaluated performance on a subset of the LRS3-TED test set filtered according to the same protocol used to construct LSVSR, by removing instances with larger face angles and shorter clips (*Filtered Test*). Second, we tested on the full unfiltered test set (*Full Test*). In both cases, V2P outperforms TM-seq2seq, achieving WERs of $47.0 \pm 1.6$ and $55.1 \pm 0.9$ respectively. This shows that our approach is able to generalize well, achieving state-of-the-art performance on datasets, with different conditions, on which it was not trained.

## 6 CONCLUSIONS

We presented a novel, large-scale visual speech recognition system. Our system consists of a data processing pipeline used to construct a vast dataset—an order of magnitude greater than all previous approaches both in terms of vocabulary and the sheer number of example sequences. We described a scalable model for producing phoneme and word sequences from processed video clips that is capable of nearly halving the error rate of the previous state-of-the-art methods on this dataset, and achieving a new state-of-the-art in a dataset presented contemporaneously with this work. The combination of methods in this work represents a significant improvement in lipreading performance, a technology which can enhance automatic speech recognition systems, and which has enormous potential to improve the lives of speech impaired patients worldwide.

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

## A    MEDICAL APPLICATIONS

As a consequence of injury or disease and its associated treatment, millions of people worldwide have communication problems preventing them from generating sound. As hearing aids and cochlear transplants have transformed the lives of people with hearing loss, there is potential for lip reading technology to provide alternative communication strategies for people who have lost their voice.

**Aphonia** is the inability to produce voiced sound. It may result from injury, paralysis, removal or other disorders of the larynx. Common examples of primary aphonia include bilateral recurrent laryngeal nerve damage as a result of thyroidectomy *(removal of the thyroid gland and any tumour)* for thyroid cancer, laryngectomy *(surgical removal of the voice box)* for laryngeal cancers, or tracheostomy *(the creation of an alternate airway in the neck bypassing the voicebox)*.

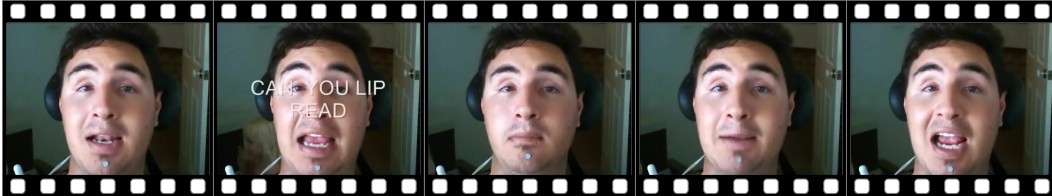

Figure 6: V2P could be helpful for performing silent speech recognition for those with aphonia[2].

**Dysphonia** is difficulty in speaking due to a physical disorder of the mouth, tongue, throat, or vocal cords. Unlike aphonia, patients retain some ability to speak. For example, in Spasmodic dysphonia, a disorder in which the laryngeal muscles go into periods of spasm, patients experience breaks or interruptions in the voice, often every few sentences, which can make a person difficult to understand.

We see this work having potential medical applications for patients with aphonia or dysphonia in at least two distinct settings. Firstly, an acute care setting (i.e. a hospital with an emergency room and an intensive care unit), patients frequently undergo elective (planned) or emergency (unplanned) procedures (e.g. Tracheostomy) which may result in aphonia or dysphonia. In the U.S. 103,925 tracheostomies were performed in 2014, resulting in an average hospital stay of 29 days (HCUPnet, 2014). Similarly, in England and Wales 15,000 tracheostomies are performed each year The Health Foundation (2014).

Where these procedures are unplanned, there is often no time or opportunity to psychologically prepare the patient for their loss of voice, or to teach the patient alternative communication strategies. Some conditions that necessitate tracheotomy, such as high spinal cord injuries, also affect limb function, further hampering alternative communication methods such as writing.

Even where procedures are planned, such as for head and neck cancers, despite preparation of the patient through consultation with a speech and language therapist, many patients find their loss of voice highly frustrating especially in the immediate post-operative period.

Secondly, where surgery has left these patients cancer-free, they may live for many years, even decades without the ability to speak effectively, in these patients we can envisage that they may use this technology in the community, after discharge from hospital. While some patients may either have tracheotomy reversed, or adapt to speaking via a voice prosthesis, electro-larynx or esophageal speech, many patients do not achieve functional spoken communication. Even in those who achieve good face-to-face spoken communication, few laryngectomy patients can communicate effectively on the telephone, and face the frequent frustration of being hung-up on by call centres and others who do not know them.

**Acute care applications.** It is widely acknowledged that patients with communication disabilities, including speech impairment or aphonia can pose significant challenges in the clinical environment, especially in acute care settings, leading to potentially poorer quality of care (Morris & Kho, 2014). While some patients will be aware prior to surgery that they may wake up unable to speak, for many patients in the acute setting (e.g. Cervical Spinal Cord Injury, sudden airway obstruction) who wake

---

[2]https://www.youtube.com/watch?v=FwOLHtHrVbc

up following an unplanned tracheotomy, their sudden inability to communicate can be phenomenally distressing.

**Community applications.** Patients who are discharged from hospital without the ability to speak, or with poor speech quality, face a multitude of challenges in day-to-day life which limits their independence, social functioning and ability to seek employment.

We hypothesize that the application of technology capable of lip-reading individuals with the ability to move their facial muscles, but without the ability to speak audibly could significantly improve quality of life for these patients. Where the application of this technology improves the person's ability to communicate over the telephone, it would enhance not only their social interactions, but also their ability to work effectively in jobs that require speaking over the phone.

Finally, in patients who are neither able to speak, nor to move their arms, this technology could represent a step-change in terms of the speed at which they can communicate, as compared to eye-tracking or facial muscle based approaches in use today.

## B  PHONEME CONFUSION MATRIX

To compute the confusion matrix and the insertion/deletion chart shown in the main text in Figure 4, we first compute the edit distance dynamic programming matrix between each predicted sequence of phonemes and the corresponding ground-truth. Then, a backtrace through this matrix gives an alignment of the two sequences, consisting of edit operations paired with positions in the prediction/ground-truth sequences.

Counting the correct phonemes and the substitutions yields the confusion matrix Figure 7. The reader can note that the diagonal is strongly dominant. A few groups are commonly confused as expected due to their visual similarity, such as {/d/, /n/, /t/}, and to a lesser extent {/b/, /p/}.

Counting insertions/deletions yields Figure 4 in the main text, showing which phonemes are most commonly omitted (deleted), or less frequently, erroneously inserted.

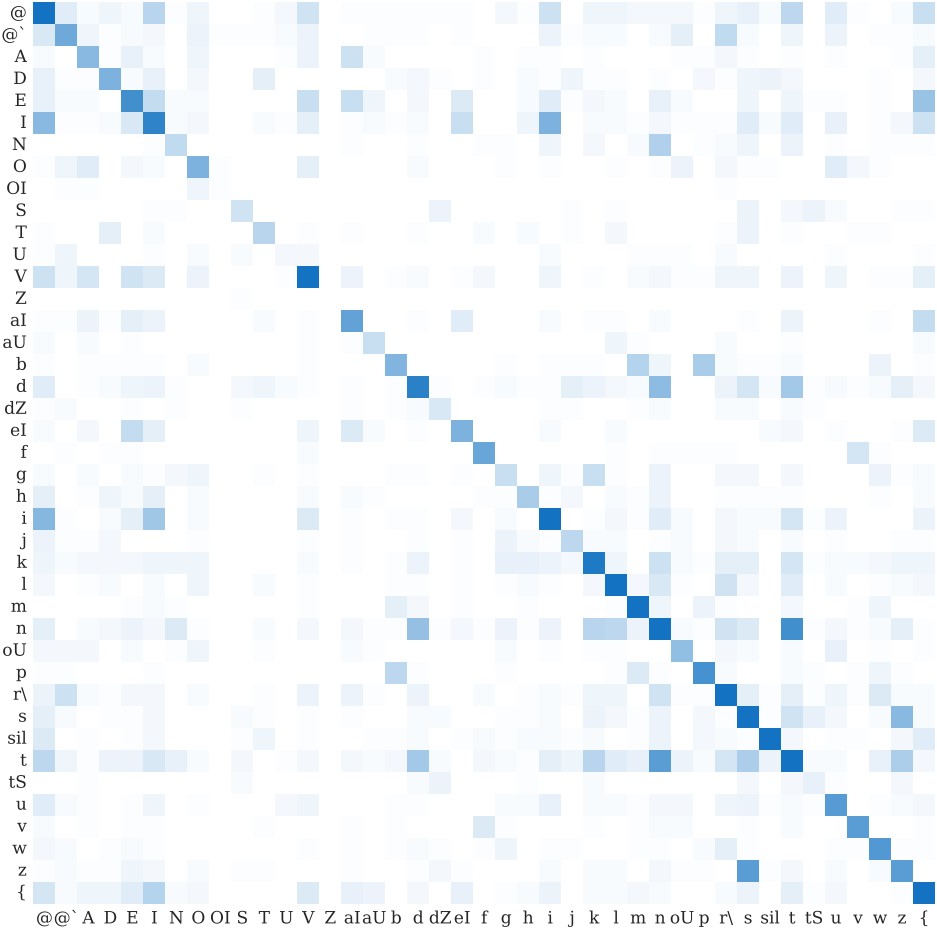

Figure 7: Phoneme confusion matrix for V2P, estimated by computing the edit distance alignment between each predicted sequence of phonemes and the corresponding ground-truth, and counting the correct phonemes and the substitutions. The diagonal values are scaled downwards to de-emphasize the correct phonemes. Blue indicates more substitutions occurred.

## C    PHONEME DELETIONS, INSERTIONS, SUBSTITUTIONS

Using the same method for computing the edit distance as described in Appendix B, Table 3 shows the absolute numbers of insertions/deletions/substitutions for the Audio-Ph and the V2P model. As it can be seen, the percentage of insertions and substitutions is substantially higher when using the visual channel (V2P) compared to the audio channel (Audio-Ph). Different phonemes can be visually identical and the information to disambiguate them is missing from the visual channel, this may result to a higher insertions and substitutions rate.

Table 3: Audio-ph and V2P phoneme deletions, insertions, and substitutions.

|               | Audio-ph |           | V2P   |           |
|---------------|----------|-----------|-------|-----------|
| Deletions     | 849      | (38.9%)   | 1366  | (10.2%)   |
| Insertions    | 1245     | (26.5%)   | 5378  | (40.0%)   |
| Substitutions | 1109     | (34.6%)   | 6697  | (49.8%)   |
| Total         | 3203     |           | 13441 |           |

## D    FACE ROTATION VS. PERFORMANCE HEATMAP

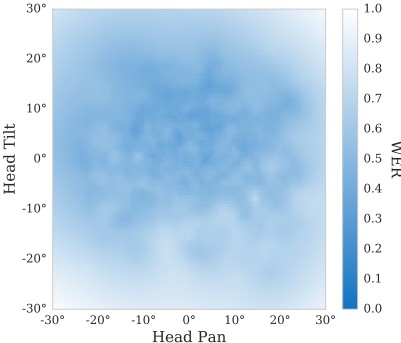

Figure 8: Heatmap showing the performance of V2P on different head rotations. Tilt and pan axes are in degrees. As shown, it performs similarly at all pan and tilt angles in $[-30°, 30°]$, the range at which it was trained.

# E  SALIENCY MAPS

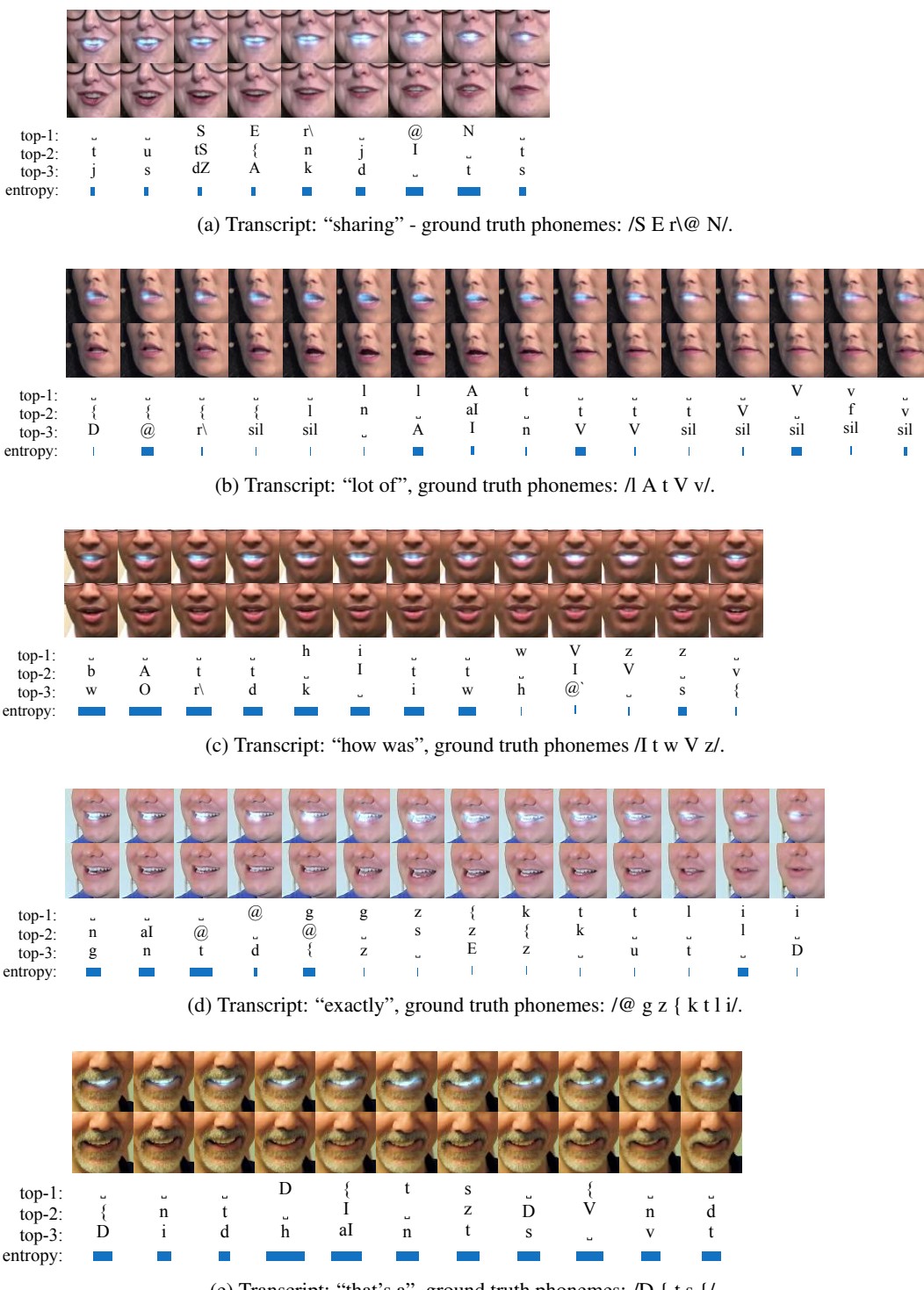

(a) Transcript: "sharing" - ground truth phonemes: /S E r\@ N/.

(b) Transcript: "lot of", ground truth phonemes: /l A t V v/.

(c) Transcript: "how was", ground truth phonemes /I t w V z/.

(d) Transcript: "exactly", ground truth phonemes: /@ g z { k t l i/.

(e) Transcript: "that's a", ground truth phonemes: /D { t s {/.

Figure 9: Saliency maps, the top-3 predictions of each frame and the ground truth phonemes.

## F  V2P-SYNC ARCHITECTURE

The V2P-Sync networks in Tables 4 and 5 are optimized using a batch size of 128, batch normalization, and Adam (Kingma & Ba, 2014) with a learning rate of $10^{-4}$ and default hyperparameters: first and second momentum coefficients 0.9 and 0.999 respectively, and $\epsilon = 10^{-8}$ for numerical stability.

Table 4: V2P-Sync video embedding neural network architecture.

| Layer | Filter size | Stride | Output channels | Input |
|---|---|---|---|---|
| conv1 | $3 \times 3 \times 3$ | $1 \times 2 \times 2$ | 16 | $9 \times 128 \times 128 \times 1$ |
| pool1 | $1 \times 2 \times 2$ | $1 \times 2 \times 2$ | | $7 \times 63 \times 63 \times 16$ |
| conv2 | $3 \times 3 \times 3$ | $1 \times 1 \times 1$ | 32 | $7 \times 31 \times 31 \times 16$ |
| pool2 | $1 \times 2 \times 2$ | $1 \times 2 \times 2$ | | $5 \times 29 \times 29 \times 32$ |
| conv3 | $3 \times 3 \times 3$ | $1 \times 1 \times 1$ | 64 | $5 \times 14 \times 14 \times 32$ |
| pool3 | $1 \times 2 \times 2$ | $1 \times 2 \times 2$ | | $3 \times 12 \times 12 \times 64$ |
| conv4 | $3 \times 3 \times 3$ | $1 \times 1 \times 1$ | 128 | $3 \times 6 \times 6 \times 64$ |
| pool4 | $1 \times 2 \times 2$ | $1 \times 2 \times 2$ | | $1 \times 4 \times 4 \times 128$ |
| fc5 | | $1 \times 1 \times 1$ | 256 | 512 |
| fc6 | | $1 \times 1 \times 1$ | 64 | 256 |

Table 5: V2P-Sync audio embedding neural network architecture.

| Layer | Support | Stride | Filters | Input |
|---|---|---|---|---|
| conv1 | $3 \times 5$ | $1 \times 1$ | 16 | $16 \times 40 \times 1$ |
| pool1 | $1 \times 2$ | $1 \times 2$ | $14 \times 36 \times 16$ | |
| conv2 | $3 \times 4$ | $1 \times 1$ | 32 | $14 \times 36 \times 16$ |
| conv3 | $3 \times 4$ | $1 \times 1$ | 32 | $12 \times 15 \times 32$ |
| pool3 | $1 \times 2$ | $1 \times 2$ | $10 \times 12 \times 32$ | |
| conv4 | $3 \times 3$ | $1 \times 1$ | 64 | $10 \times 6 \times 32$ |
| conv5 | $3 \times 3$ | $1 \times 1$ | 64 | $8 \times 4 \times 64$ |
| conv6 | $3 \times 2$ | $1 \times 1$ | 128 | $6 \times 2 \times 64$ |
| fc7 | | $1 \times 1$ | 256 | 512 |
| fc8 | | $1 \times 1$ | 64 | 256 |

## G   V2P ARCHITECTURE

The network architecture is optimized using Adam (Kingma & Ba, 2014) with a learning rate of $10^{-4}$ and default hyperparameters: first and second momentum coefficients 0.9 and 0.999 respectively, and $\epsilon = 10^{-8}$ for numerical stability. Furthermore, to accelerate learning, a curriculum schedule limits the video duration, starting from 2 seconds and gradually increasing to a maximum length of 12 seconds over 200,000 training steps. Finally, image transformations are also applied to augment the image frames to help improve invariance to filming conditions. This is accomplished by first randomly mirroring the videos horizontally, followed by random changes to brightness, contrast, saturation, and hue.

Table 6: V2P architecture details.

| Layer | Filter size | Stride | Output channels | Input |
|---|---|---|---|---|
| conv1 | $3 \times 3 \times 3$ | $1 \times 2 \times 2$ | 64 | T $\times 128 \times 128 \times 3$ |
| pool1 | $1 \times 2 \times 2$ | $1 \times 2 \times 2$ | | T $\times 63 \times 63 \times 64$ |
| conv2 | $3 \times 3 \times 3$ | $1 \times 1 \times 1$ | 128 | T $\times 31 \times 31 \times 64$ |
| pool2 | $1 \times 2 \times 2$ | $1 \times 2 \times 2$ | | T $\times 29 \times 29 \times 128$ |
| conv3 | $3 \times 3 \times 3$ | $1 \times 1 \times 1$ | 256 | T $\times 14 \times 14 \times 128$ |
| pool3 | $1 \times 2 \times 2$ | $1 \times 2 \times 2$ | | T $\times 12 \times 12 \times 256$ |
| conv4 | $3 \times 3 \times 3$ | $1 \times 1 \times 1$ | 512 | T $\times 6 \times 6 \times 256$ |
| conv5 | $3 \times 3 \times 3$ | $1 \times 1 \times 1$ | 512 | T $\times 4 \times 4 \times 512$ |
| pool5 | $1 \times 2 \times 2$ | $1 \times 1 \times 1$ | | T $\times 2 \times 2 \times 512$ |
| bilstm6 | | | $768 \times 2$ | T $\times 512$ |
| bilstm7 | | | $768 \times 2$ | T $\times 1536$ |
| bilstm8 | | | $768 \times 2$ | T $\times 1536$ |
| fc9 | | | 768 | T $\times 1536$ |
| fc10 | | | $41 + 1$ | T $\times 768$ |

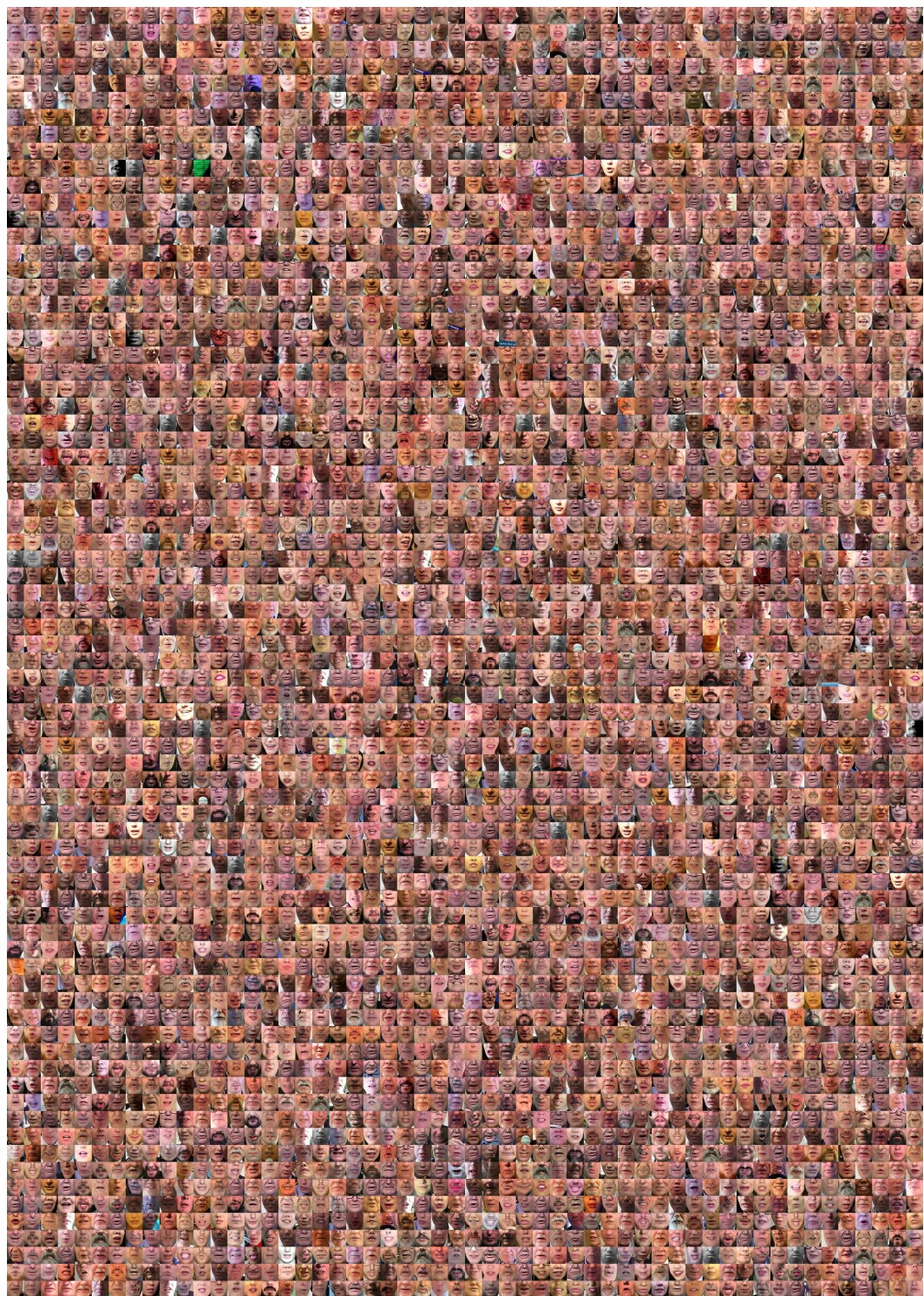

Figure 10: Random sample of test-set lip images from LSVSR. This illustrates the substantial diversity in our dataset.

