# OpenReview forum: " Large-Scale Visual Speech Recognition"
_ICLR.cc/2019/Conference_

### Official Review · AnonReviewer3 · 2018-11-01
**review of the paper**

**Rating:** 9
**Confidence:** 4

**Review:**

This is a good paper. First of all, it presents a large-scale corpus for visual speech recognition. Second, it demonstrates a visual speech recognition system based on open-vocabulary that gives the state-of-the-art recognition accuracy.  The paper is very well written and all the technical details are clearly laid out.  I, for one, would like to thank the authors for this meticulous work to the community.   This is by far the largest dataset and the most impressive performance for VSR I have even seen in the ASR/VSR community.  I enjoyed reading this paper.

I extend this review based on the replies.  One of the arguments is that the work presented in this paper is a great success in engineering but it lacks technical novelty and therefore can not be accepted by the conference, which I think otherwise.  First of all, the authors put together a very detailed and carefully designed technical pipeline for creating a very large visual speech recognition dataset, which is a valuable contribution to be community.  (I assumed that the databset will become available to the community when reviewing the paper, which turned out not to be totally accurate. My apologies.  I do hope the dataset will be made public.  This is a major reason I gave a high score.)  Second,  the authors have built systems that give the state-of-the-art performance on visual speech recognition. Although the models and architectures are already out there,  the impressive performance itself is an impact to the field.  This is not simply achieved by piling in a large amount of data (although it does play a role). This is a system paper but its impact and its performance should at least get it in to the conference.

---

> ### Author Response · Authors · 2018-11-06
> **Authors' Response**
>
> Thank you for your encouraging feedback.

---

### Official Review · AnonReviewer1 · 2018-11-03
**Engineering Marvel**

**Rating:** 3
**Confidence:** 5

**Review:**

The paper presents a large-scale lipreading system - no surprises there. This is good work and probably the strongest general purpose lip-reading system out there at this time, but i don't see both the work and the paper as a good fit for ICLR.

The authors take a large corpus of YouTube videos (on which Google has already trained direct acoustics-to-word speech recognizers, and which is manually transcribed), filter it, and extract regions that can be used for lipreading. They then describe a scalable preprocessing, and train a phone-based acoustic model using CTC. They seem to be using the (Miao et al., 2015) and Google WFST based decoding framework, and achieve a word error rate of ca 40%. That is impressive, but I don't see any novelty here, and the paper is full of contradictions, and leaves some important open questions:

- the authors argue for "phonemes and ctc", and no speech person would disagree with them; in fact (Miao et al., 2015) and many other papers show that the WERs with a good phoneme based dictionary in English are lower than with a character based model. it's just easier if one does not need a dictionary.
- why are the authors not using a viseme dictionary, or map their phoneme dictionary to a viseme dictionary. In visual space, their own "homonym" argument applies, too, and "mop" (or "mom") and "pop" should be mapped to the same "viseme" sequence - and the resulting uncertainty should be handled by the decoder, and not the classifier.
- how did the authors generate the one million word phoneme vocabulary? even google used around 100,000k words in their whole-word experiments, if i remember correctly? what happens if the authros reduce the vocabulary? could you provide some error analysis or at least deletions/ insertions/ substitutiosn, and compare them against an audio system?
- LipNet and the proposed architecture seem to be very similar - maybe you could provide some insight into which changes made the biggest difference?
- is the data going to be available?
- what is a "production-level speech decoder"? how come your model "is the first to combine a deep learning-based phoneme recognition model with production-grade word-level decoding techniques" if Google does essentially the same ("in production")?
- in Section 1, you say that "by design, the trained model only performs well when videos are shot at specific angles when a subject is facing the camera, [...] It does not perform well in other contexts". in Section 5, you demonstrate the "generalization power of our V2P approach"and find that it "is able to generalize well" - please clarify
- "speech impaired patients" often have non-canonical articulation, the proposed system may not work well for them
- it would be interesting to also know the absolute levels of insertions/ deletions/ substitutions for words and/ or phonemes, and for the audio only and visual systems, to be able to diagnose what the problems are.
- finally, Figure 10 is really hard to view - i'd be happy to be shown fewer faces, the main message is that the quality of the face detection is really good?

---

> ### Author Response · Authors · 2018-11-06
> **Authors' Response 2**
>
> > could you provide some error analysis or at least deletions/ insertions/ substitutiosn, and compare them against an audio system?
>
> We provide heatmaps with deletions / insertions / substitutions in Figures 4 and 7 for V2P, and we could easily provide similar results for our audio baseline. Thanks for the suggestion.
>
> > LipNet and the proposed architecture seem to be very similar - maybe you could provide some insight into which changes made the biggest difference?
>
> Overall we found that 1) introducing stabilization in the processing pipeline, changing the 2) network size, 3) depth, 4) replacing dropout with group normalization, and 5) working in phoneme level with a decoder pipeline were the key components of our performance, an ablation for each is shown in Table 1. We will ensure the differences are better emphasized in the main text.
>
> > is the data going to be available?
>
> We are very interested in publishing our dataset / video timestamps and we are investigating possible ways to do so. We will only do so provided there are no privacy or security concerns. The massive scale and importance of this dataset demands responsible use.
>
> > what is a "production-level speech decoder"? how come your model "is the first to combine a deep learning-based phoneme recognition model with production-grade word-level decoding techniques" if Google does essentially the same ("in production")?
>
> We mean specifically in the context of visual speech recognition. That is, the phoneme recognition model takes videos as input, not audio. Thank you, we will clarify it in our next update.
>
> > in Section 1, you say that "by design, the trained model only performs well when videos are shot at specific angles when a subject is facing the camera, [...] It does not perform well in other contexts". in Section 5, you demonstrate the "generalization power of our V2P approach"and find that it "is able to generalize well" - please clarify
>
> We apologize for the confusion. We will be more precise in the next version. The precise facts are as reported: In the TED experiments we examined the performance of speaker angles outside our training set and, as shown in Table 2, outside this range the performance dropped by 8 WER, but indeed our model still outperforms TM-seq2seq. Thus, there is a drop in performance and this performance will eventually drop to zero if the lips stop appearing, but our approach is better at generalizing than other approaches.
>
> > "speech impaired patients" often have non-canonical articulation, the proposed system may not work well for them
>
> Correct, hyperarticulation is a case where our proposed system would not work well. We are working closely with specialists for identifying the cases we can help. Our preliminary investigation can be found in Appendix A. We expect that patients who spoke normally for their whole lives but only recently lost the ability to produce sound will retain mostly normal articulation. We have conducted some tests and the results are positive, and as soon as we obtain proper approval, we will release these.
>
> > it would be interesting to also know the absolute levels of insertions/ deletions/ substitutions for words and/ or phonemes, and for the audio only and visual systems, to be able to diagnose what the problems are.
>
> A table with absolute values wouldn't easily fit in the PDF, but we can include these as additional supplementary material.
>
> > finally, Figure 10 is really hard to view - i'd be happy to be shown fewer faces, the main message is that the quality of the face detection is really good?
>
> The point of this picture was to show the diversity of the LSVSR dataset. Thanks, we will add a note to this effect.
>
>
> References:
> [1] Chung, Joon Son, et al. "Lip Reading Sentences in the Wild." CVPR. 2017.
>
> [2] Bear, Helen L., and Richard Harvey. "Phoneme-to-viseme mappings: the good, the bad, and the ugly." Speech Communication 95 (2017): 40-67.
>
> [3] Fisher, Cletus G. "Confusions among visually perceived consonants." Journal of Speech, Language, and Hearing Research 11.4 (1968): 796-804.
>
> [4] Hazen, Timothy J., et al. "A segment-based audio-visual speech recognizer: Data collection, development, and initial experiments." Proceedings of the 6th international conference on Multimodal interfaces. ACM, 2004.

---

> ### Author Response · Authors · 2018-11-06
> **Authors' Response 1**
>
> > The paper presents a large-scale lipreading system - no surprises there. This is good work and probably the strongest general purpose lip-reading system out there at this time, but i don't see both the work and the paper as a good fit for ICLR.
>
> Thank you. We feel it is a good fit for various reasons. First, this work is about an important application of deep neural networks, just like audio speech recognition, image classification and translation, where results and execution have mattered and partly led to the success of this conference. Second, our work  demonstrates scaling to massive video datasets, a frontier of great interest. Third, it demonstrates some reasons for considering dynamic programming approaches (CTC here) over seq2seq attention models, which are more popular in this community. Finally, it provides ablations revealing the importance of different modules in constructing large-scale neural network architectures.
>
> > The authors argue for "phonemes and ctc", and no speech person would disagree with them; in fact (Miao et al., 2015) and many other papers show that the WERs with a good phoneme based dictionary in English are lower than with a character based model. it's just easier if one does not need a dictionary.
>
> While this might have been known for audio ASR, it certainly was not the case in visual ASR; the topic of this paper. In fact, the previous state-of-the-art in visual speech recognition was a seq2seq character-based model [1]. For this reason, we compared approaches and showed that using "phonemes and ctc" works better than characters and seq2seq in this domain.
>
> > why are the authors not using a viseme dictionary, or map their phoneme dictionary to a viseme dictionary. In visual space, their own "homonym" argument applies, too, and "mop" (or "mom") and "pop" should be mapped to the same "viseme" sequence - and the resulting uncertainty should be handled by the decoder, and not the classifier.
>
> Recent literature shows that there is no optimal viseme mapping that can generalize to all individuals [2]; that work compares 15 different mappings and our preliminary work in the same direction shows similar trends. For example /m/ and /p/ in "mop" and "pop" don't belong to the same viseme group according to Fisher et al. [3] or Hazen et al. [4], but they do in several other mappings. In this setting, using the phonemes, we sidestep the need to choose a single viseme mapping, and instead allow the uncertainty between inherently similar/identical phonemes to be encoded in the predictive distribution of the neural network directly. Further, the language model is subsequently able to manage this resulting uncertainty by mapping sequences of phoneme distributions to sequences of words, and we are able to leverage the fact that we can build very good language models.
>
> > how did the authors generate the one million word phoneme vocabulary? even google used around 100,000k words in their whole-word experiments, if i remember correctly? what happens if the authros reduce the vocabulary?
>
> We use almost completely raw text from YT transcripts, which results in a much wider variety of words, acronyms, numbers, currency amounts, etc. than one would normally expect from cleaned text. Given the words in the training set, pronunciations are generated by a dictionary and falling back to a grapheme-to-phoneme model. We haven't tried experiments on reducing the vocabulary, but we expect it to have little effect on WER as long as it captures the probability mass. As expected WER would drop in a limited domain with a restricted vocabulary.

---

### Official Review · AnonReviewer2 · 2018-11-06
**nice data collection but no technical contribution**

**Rating:** 4
**Confidence:** 4

**Review:**

The paper presents a non-trivial data processing pipeline, a large data set, and a system based on CTC and FSTs for automatic lipreading from videos.

The review of the previous work is comprehensive. The authors are also awared of the state of the art in speech recognition, a highly related task.

The collection of the data set is definitely a contribution, but other than that, the technical novelty is scarce, since all of the techniques have been proposed either in lipreading from video or in speech recognition.

The numbers in Table 1 are impressive, but it is hard to tell where the improvement is coming from. It is worth running a few more experiments a) with the label set fixed while changing the network architecture b) with the network architecture fixed while changing the label set c) with the network and the label set fixed while changing dropout or group normalization. seq2seq is an odd child in this case, because you cannot really compare it to other settings.

The result in Table 2 is also impressive, but it would be nice to have the proposed system trained on LRS3-TED and compare against TM-seq2seq.

It is generally a consensus that a large model paired with a large amount of data gives you improvement, and this type of improvement is not considered a contribution. It is then the authors' responsibility to have a comprehensive experiments showing that the improvement is not just due to having a larger model and more data.

Here are some minor details:

p.6.

note that there must be a blank between the 'e' characters to avoid collapsing ...
--> this is actually not true, at least not in the original CTC formulation, where removing the duplicates and blanks have to be done in that order.

To explain why modeling characters with CTC is problematic, ...
--> this argument is not theoretically sound, so the question is does this happen in practice? the loss only measures at the independence level, but this doesn't prohibit the network to learn dependencies before the loss.

---

> ### Author Response · Authors · 2018-11-07
> **Authors' Response 2**
>
> > The result in Table 2 is also impressive, but it would be nice to have the proposed system trained on LRS3-TED and compare against TM-seq2seq.
> > It is generally a consensus that a large model paired with a large amount of data gives you improvement, and this type of improvement is not considered a contribution. It is then the authors' responsibility to have a comprehensive experiments showing that the improvement is not just due to having a larger model and more data.
>
> Thank you. As our model takes about a month to train even with 64 GPUs, and the LRS3-TED dataset was released less than a month before the paper submission deadline, this was deemed infeasible. We think our ablations (Table 1) make it very clear that a large model is not enough --- we explicitly controlled for this, and found that one needs a well designed large model. The point of testing on the smaller LRS3-TED dataset was solely to illustrate that our model beats the state-of-the-art there even without being trained there --- that is, the objective of the experiment was to show that our model generalizes well to other data.
>
> > note that there must be a blank between the 'e' characters to avoid collapsing ...
> --> this is actually not true, at least not in the original CTC formulation, where removing the duplicates and blanks have to be done in that order.
>
> In the original CTC formulation, indeed removal of duplicates happens before the removal of blanks. So in our example and the original CTC paper's 'B' function, B(be#ee) = bee, where # denotes blank. We will reword that sentence for clarity in the next update.
>
> > To explain why modeling characters with CTC is problematic, ...
> --> this argument is not theoretically sound, so the question is does this happen in practice? the loss only measures at the independence level, but this doesn't prohibit the network to learn dependencies before the loss.
>
> This argument is indeed backed up by our empirical findings. While the network could learn dependencies before the loss, this would hamper its ability to represent the uncertainty that is inherent in visual speech recognition due to phonemes that appear visually similar (visemes). In the case of audio ASR on clean audio, this argument is indeed less applicable.

---

> ### Author Response · Authors · 2018-11-07
> **Authors' Response 1**
>
> > The paper presents a non-trivial data processing pipeline, a large data set, and a system based on CTC and FSTs for automatic lipreading from videos.
> > The review of the previous work is comprehensive. The authors are also awared of the state of the art in speech recognition, a highly related task.
> > The collection of the data set is definitely a contribution, but other than that, the technical novelty is scarce, since all of the techniques have been proposed either in lipreading from video or in speech recognition.
>
> While the individual components exist in the literature, their combination to achieve state-of-the-art visual speech recognition is unique. Further, due to the large scale and distributed nature of our training, model and dataset, the choice of components demands additional thought. For example, at this scale we could do group-norm, but not batch-norm. Many alternative design decisions which would have been sensible at small scale fail at this large scale. In this sense, we believe we are innovating and pushing an important frontier.
>
> > The numbers in Table 1 are impressive, but it is hard to tell where the improvement is coming from. It is worth running a few more experiments a) with the label set fixed while changing the network architecture b) with the network architecture fixed while changing the label set c) with the network and the label set fixed while changing dropout or group normalization. seq2seq is an odd child in this case, because you cannot really compare it to other settings.
>
> Training our model with 64 GPUs takes about 1 month.
>
> We conducted as many ablations as possible for over a year. In TABLE 1, we report what we thought were the most important ablations, which we would like to clarify as we think these already address many of your questions. First, starting with LipNet and switching from character labels to phoneme labels results in a modest gain (~3% gain in WER), and importantly we provide a section in the paper titled rationale for phonemes which discusses this label choice.
>
> Second, we increased the size of LipNet to match the size of our V2P model and this results in a decrease of  ~20% in WER for the same labels and dataset. Yet, while size matters, clever design plays the biggest role. By focusing on architectural changes, such as group norm instead of dropout and LSTM cores instead of GRUs, we are able to reduce the WER from 72.7% to 40.9% (an improvement of about 32% WER). So these architectural changes are the most important factors to achieve our final WER, and are crucial for optimising the neural net architectures to take full advantage of multi-GPU distributed systems --- as we are pushing the limits of computation, communication and storage.
>
> Third, we provide ablations to measure the contribution of the language model and alternative architectures (fully convolutional). Finally, we compare against the seq2seq architecture because it was the previous state-of-the-art in visual speech recognition.
>
> Altogether we strongly feel this is a comprehensive and thorough set of ablations that clearly tests the value of different design choices.

---

> > ### Comment · AnonReviewer2 · 2018-11-21
> > **still not convincing**
> >
> > I think we can agree that V2P is better than the architectures that the authors tried. This paper also provides a nice recipe for this task on this data set. However, the problem is that we cannot conclude anything beyond that. I did not learn any insights about solving this task, and I also did not learn anything about the representation learned for this task.
> >
> > > Further, due to the large scale and distributed nature of our training, model and dataset, the choice of components demands additional thought. For example, at this scale we could do group-norm, but not batch-norm.
> >
> > I believe there are intuitions behind the design choices. It would be very useful to state them as claims and hypotheses and use experiments to confirm the intuition. Having said that, some of the changes, such as favoring LSTMs over GRUs, seem arbitrary.
> >
> > > Many alternative design decisions which would have been sensible at small scale fail at this large scale.
> >
> > Yes, this is not a surprising fact. Removing dropout is rather obvious when you have more data. On the other hand, I do not see why GRUs would work for small data sets and fail for large ones.
> >
> > > We conducted as many ablations as possible for over a year. In TABLE 1, we report what we thought were the most important ablations, which we would like to clarify as we think these already address many of your questions. First, starting with LipNet and switching from character labels to phoneme labels results in a modest gain (~3% gain in WER), and importantly we provide a section in the paper titled rationale for phonemes which discusses this label choice.
> >
> > I agree. I should emphasize that the LipNet experiments are better than the others because the four rows are control experiments of each other. However, not much can be concluded by comparing Baseline-LipNet-Large-Ph to V2P.
> >
> > > Second, we increased the size of LipNet to match the size of our V2P model and this results in a decrease of  ~20% in WER for the same labels and dataset. Yet, while size matters, clever design plays the biggest role. By focusing on architectural changes, such as group norm instead of dropout and LSTM cores instead of GRUs, we are able to reduce the WER from 72.7% to 40.9% (an improvement of about 32% WER). So these architectural changes are the most important factors to achieve our final WER, and are crucial for optimizing the neural net architectures to take full advantage of multi-GPU distributed systems --- as we are pushing the limits of computation, communication and storage.
> >
> > These architectural changes are not novel. I also do not see how LSTMs are better than GRUs at utilizing multiple GPUs.
> >
> > > Third, we provide ablations to measure the contribution of the language model and alternative architectures (fully convolutional). Finally, we compare against the seq2seq architecture because it was the previous state-of-the-art in visual speech recognition.
> >
> > Again, I get the fact that V2P is better. Adding an LM is not something novel. I also do not understand the point of using a fully convolutional network. I guess it's about the amount of GPU utilization, but I didn't see comprehensive experiments on this either.

---

### Public Comment · (anonymous) · 2018-11-08
**Side-view faces and pipeline contribution**

Interesting engineering work. Could you please elaborate on the following questions?

The samples in Fig. 10 are all looking (almost) towards the camera. What happens to the method for side-view of the face? Several lines of work in the vision community focus on that, e.g. face recognition. Is this a limitation of the dataset?

The authors claim in a previous comment that this pipeline is a significant contribution, however significant part of this pipeline has been used in previous works (e.g. https://arxiv.org/pdf/1705.02966.pdf or its more recent extensions).

A major point of the paper is the engineering pipeline. Can the authors explain what other methods have they tried? A similar work submitted in this conference describes in details such efforts:
https://openreview.net/pdf?id=B1xsqj09Fm (appendix C and E).

---

> ### Author Response · Authors · 2018-11-08
> **Authors' Response**
>
> > Interesting engineering work. Could you please elaborate on the following questions?
>
> Thank you. We will try to do our best to answer your questions.
>
> > The samples in Fig. 10 are all looking (almost) towards the camera. What happens to the method for side-view of the face? Several lines of work in the vision community focus on that, e.g. face recognition. Is this a limitation of the dataset?
>
> The choice of +/- 30 degree face views was driven by our desire to focus on apps with good social impact, where the user has an incentive to look at the camera. Our intention is to purposely design tools to help people with speech impairments; 100,000s could benefit. We also recommend you examine the last paragraph of Section 5.1 and Table 2. There, our experiments show that even though our method was trained on faces at angles of +/- 30 degrees it outperforms the state of the art at angles of +/- 90 degrees on a different dataset that allows for this greater variation. Table 2 has the precise numbers.
>
> > The authors claim in a previous comment that this pipeline is a significant contribution, however significant part of this pipeline has been used in previous works (e.g. https://arxiv.org/pdf/1705.02966.pdf or its more recent extensions).
>
> While this important line of work influenced our research, there are large and  important differences in our preprocessing pipelines, e.g. landmark smoothing, blurriness filtering, and extracting phoneme alignments, which differ from previous work on visual ASR and which we found crucial for obtaining a clean dataset from such a noisy source as YouTube. We will make sure to highlight these differences.
>
> We should also distinguish between the “pre-processing pipeline” used to generate the dataset and the neural network architecture used for visual ASR. In this work we also provide comparisons with the (very different) neural network architecture proposed by the team you cite (seq2seq).
>
> > A major point of the paper is the engineering pipeline. Can the authors explain what other methods have they tried? A similar work submitted in this conference describes in details such efforts:
>
> > https://openreview.net/pdf?id=B1xsqj09Fm (appendix C and E).
>
> With regards to the neural network architecture, we report a comprehensive set of the most important ablations, in Table 1, although we should note that each ablation takes about a month of computing on 64 GPUs.
>
> Thanks for drawing the connection to this other ICLR submission (Large Scale GAN) on scaling up representations and datasets. Indeed scaling up neural network representations is a very important topic in deep learning and at ICLR, as we pointed out to AnonReviewer1.
>
> Appendix C of the GAN paper touches on the difficulties associated with batch norm at scale. In this work the authors employ cross-replica Batch Norm, but in preliminary experiments we found this level of communication to be too expensive computationally. At sampling time the GAN work does make use of a GAN-specific approach which does not apply in our setting. Similar approaches could be considered in our setting, however they often trade off computational time for memory, and in our setting we are at the very limit of available memory in modern distributed GPU architectures due to the fact that video data has the extra time dimension (e.g. 360 frames for 12 seconds). As a result we found group-norm to be a very effective solution and it is one of the greatest contributors to our results (as it can be seen in Table 1). We will add further details discussing these tradeoffs to the paper.

---

> > ### Public Comment · (anonymous) · 2018-11-10
> > **Additional clarifications for engineering pipeline and dataset**
> >
> > Thanks for the quick replies. However, I feel several significant details are somehow left out of the paper (or at least the authors' replies are not clear in the paper).
> >
> > The authors mention that significant difference from previous pipelines are used. Could the authors mention those clearly in the paper? Because, several lines of work use almost the same pipeline. It cannot be as a contribution in all the works.
> >
> > In addition about reproducibility: If it is so hard to train a model, are the authors planning on releasing the code along with the trained models?
> >
> > Since the authors replied that one of their driving motivations is large scale models, shouldn't they at least share some detailed insights? The models in Table 1 are a handful. If the main novelty is an large-scale pipeline, shouldn't a thorough experimentation take place?  Obviously the authors seem to have the resources, so why not perform the proper analysis?
> >
> > The dataset seems to be the core contribution to the community from this work. Have the authors made any other implicit assumptions that a user should be aware of (apart from the degrees of face rotation)? What about ethnicities included?

---

> > > ### Author Response · Authors · 2018-11-15
> > > **Authors' Response 2**
> > >
> > > > In addition about reproducibility: If it is so hard to train a model, are the authors planning on releasing the code along with the trained models?
> > >
> > > While training the models benchmarked in the paper is computationally intensive, it is not necessarily difficult to implement our model given our description in the paper. If you feel there are architectural details of our model (V2P) missing from our paper please let us know, and we will do our best to assist.
> > >
> > > > Since the authors replied that one of their driving motivations is large scale models, shouldn't they at least share some detailed insights? The models in Table 1 are a handful. If the main novelty is an large-scale pipeline, shouldn't a thorough experimentation take place?  Obviously the authors seem to have the resources, so why not perform the proper analysis?
> > > > The dataset seems to be the core contribution to the community from this work.
> > >
> > > The models benchmarked against in Table 1 represent the state-of-the-art over the last two years. The comparisons also include important ablations, which illustrate the value of what we think are the important things to know about when training models at this large scale. For example, it becomes essential to consider alternatives to batch normalization, as also pointed out in another effort being reviewed at this conference focusing on scale ( https://openreview.net/pdf?id=B1xsqj09Fm ). We hope this shared knowledge will be useful to other practitioners in our community.
> > >
> > > For clarity, the novelty claims of the paper extend beyond the data processing pipeline or dataset, and are detailed in the introduction. These claims are empirically validated in the evaluation section.
> > >
> > > We present as many ablations as possible over months of experimentation, and again emphasize that we report what we think will be most useful to other researchers interested in large-scale experiments. We would be delighted to answer any questions in relation to this.
> > >
> > > > Have the authors made any other implicit assumptions that a user should be aware of (apart from the degrees of face rotation)? What about ethnicities included?
> > >
> > > The dataset originates from YouTube and captures the wide diversity of people in that medium. Figure 10 illustrates random samples from CC videos of our dataset. As stated previously, having shared this knowledge with the research community, our current focus is on developing accessibility apps for speech and hearing impaired people. As such, we obviously need a method that works for everyone.

---

> > > ### Author Response · Authors · 2018-11-15
> > > **Authors' Response 1**
> > >
> > > > Thanks for the quick replies. However, I feel several significant details are somehow left out of the paper (or at least the authors' replies are not clear in the paper).
> > >
> > > Thank you for your feedback, we will do our best to try to clarify your questions next.
> > >
> > > > The authors mention that significant difference from previous pipelines are used. Could the authors mention those clearly in the paper? Because, several lines of work use almost the same pipeline. It cannot be as a contribution in all the works.
> > >
> > > We intend to incorporate the clarifications, mentioned in our previous reply and expanded below, in the revised version of the paper. As mentioned before, while we agree that the important line of work of Chung et. al., 2016 (https://arxiv.org/pdf/1611.05358.pdf) has influenced our research, there are some important differences between our preprocessing pipeline and previous literature. In particular, our pipeline has the following key differences:
> > >
> > > a) Landmark smoothing. The outputs of the resulting landmark positions from the face tracker module are smoothed using a temporal Gaussian kernel. Intuitively, this simplifies learning filters for the 3D convolution layers by reducing spatiotemporal noise. Empirically, our preliminary studies showed smoothing was crucial for achieving optimal performance. Chung et. al. don't perform this smoothing step.
> > >
> > > b) Quality filtering. While we limit the minimum distance between the eyes to 80px, which allows us to keep only high resolution samples, we have additionally found that using the variance of Laplacian of each frame to be a very effective filter for blurriness detection in videos where the resolution is still high. This is something novel in the visual ASR literature. Chung et al. do not perform this type of filtering as their videos are limited to a standard format originating from the same source and our dataset is much more varied.
> > >
> > > c) Low computational cost speaking filter. Processing ~16 years of video can be computationally intensive, this module had a crucial role in the performance of our preprocessing pipeline. Using the extracted and smoothed landmarks, minor lip movements and non speaking faces are discarded using a threshold filter on the standard deviation of the mouth openness. This classifier has very low computational cost, but high recall, e.g. voice-overs are not handled. This component is distinct from the speaking filter of Chung et al. but follows the same intuition. Arguably, in both cases the classifiers are noisy, and as noted above have high recall. The noise in this classifier may be slightly ameliorated by our use of the landmark smoothing, however this is a point that deserves further study.
> > >
> > > d) Speaking classifier. V2P-Sync is our proposed architecture for an audio-video synchronisation classifier. We have found that V2P-Sync performs better compared to earlier work by Chung & Zisserman (2016b) and Torfi et al. (2017). The key difference are the following: 1) compared to Chung & Zisserman our work uses a 3D convolutions as from our V2P evaluation we've found they are much more applicable to video, 2) Torfi et al. do use 3D convolutions and a softmax classifier but doesn't use landmark smoothing, view canonicalisation and the inputs are lower resolution (100x60 vs 128x128). In practice we found that training a max margin classifier was easier. More details on the architecture of V2P-sync can be found in the Appendix section of our paper.
> > >
> > > e) Phonemes. The aligned phoneme sequences are obtained from the transcripts via a standard forced alignment approach using a lexicon with multiple pronunciations. While the use of phonemes is a crucial part of our model and architecture, this is also something that must be extracted by the pre-processing pipeline, and to the best of our knowledge this is the first work to make use of a phoneme-level end-to-end trained model for visual ASR.

---

### Public Comment · (anonymous) · 2018-11-10
**Gap between size of train and test sets**

The authors mention that the model is trained on "3,886 hours of video" but only evaluated "on a held-out test set roughly 37 minutes long". Is there a reason for this gap? What would happen if a less radical train-test split is used? Would the results still be as good? Otherwise it is hard to tell whether the improvement comes from the model or the data, as AnnonReviewer2 pointed out.

---

> ### Author Response · Authors · 2018-11-15
> **Authors' Response**
>
> This is an excellent observation. The current split was a consequence of how the data was generated and filtered. We fully agree that the test set could be larger and are working on this. However, we anticipated this and for this reason we also tested on the LRS3-TED test set, and report the results in the paper. In this out-of-sample test, the proposed approach outperformed the state-of-the-art method trained on LRS3-TED, thus providing us with good evidence of its generalization capabilities.

---

### Author Response · Authors · 2018-11-26
**A revision is submitted following the feedback from all reviewers and commenters.**

This paper presents the first production-scale solution to lipreading. Up to this point this had only been a research endeavor or a sci-fi vision. This paper has shown that this vision is possible in practice for the first time, and has done so by providing SOTA results that far surpass any previous efforts in the literature and the key baselines. The  ablations extended the models, which were the SOTA for the last two years, to make them comparable at the same large scale. The ablations also showed which engineering decisions mattered the most.

Whether this is seen as novel or a good fit to ICLR is something we cannot prove objectively. In this we are at a loss, with review scores ranging from 3 (clear rejection - despite the reviewer stating in his title that this is an Engineering Marvel) to 9 (Top 15% of accepted papers).

We thank the reviewers and anonymous commentators for their questions and insights. We have done our best to answer the technical questions on this site, and we have updated the paper with more details in light of the feedback we obtained. We hope that as a result we have all contributed to make the approach more clear for everyone.

Thank you.

---

> ### Comment · AnonReviewer2 · 2018-11-28
> **thanks for the great effort**
>
> Thanks for the update. The revised paper reads better.
>
> I think the data set and the system are a great contribution and will have a large impact. However, I still think this paper has little technical novelty, because I did not learn anything after reading the paper except that this task is hard. The overall approach is identical to (Miao et al., 2015) and the network architecture is very similar to (Sainath et al., 2015).
>
> I agree with Reviewer 1, and find it hard to give a high score. I am more comfortable giving a high score if the paper is written in a way that focuses on the data set and the data set preparation pipeline while providing a strong baseline.
>
> EESEN: end-to-end speech recognition using deep RNN models and WFST-based decoding
> Yajie Miao, Mohammad Gowayyed, Florian Metze
> ASRU, 2015
>
> Convolutional, long short-term memory, fully connected deep neural networks
> Tara N. Sainath, Oriol Vinyals, Andrew Senior, Hasim Sak
> ICASSP, 2015

---

> ### Comment · AnonReviewer1 · 2018-11-28
> **much improved - but fundamentally still not a good fit**
>
> I want to thank the authors for the sincere efforts and express again that i think this is impressive, interesting and relevant work. The new version of the paper reads much better, and addresses some of my criticism, but the fundamental issue is still: I find it hard to get overly excited about a paper that doesn't really advance our understanding about "learning representations". There are still some places where I think my original criticism still applies, but their number has reduced.
>
> In the current version, the authors make a point that this paper is important because of (1) the large data set/ good performance, (2) the potential for helping the hard of hearing, and (3) the relevance of the image processing pipeline. This suggests that the paper could also go to a large-data and/ or engineering oriented venue, an accessibility leaning venue, or a more applied speech/ vision (maybe even multi-modal?) place.
>
> Given the current paper, I'd find it hard to identify a single or minimal set of changes that would be made to make this paper a good fit for ICLR, but I think there would need to be some insight into what and why is the model doing what it is doing, and some issues (will the data be released? why are phonemes being used and not visemes?) should be addressed.
>
> I don't envy the area chairs ...

---

> ### Comment · AnonReviewer3 · 2018-11-28
> **stay with my score**
>
> Thanks to the authors for the revision and detailed response to all the concerns/comments raised by the reviewers.  As I said, I enjoyed reading this paper which I think is a really good piece of work with a great impact to the community.  Personally, I think it is a fit to this conference and deserves to get in.  The other two reviewers, however, think otherwise. I will stay with my score and leave the decision to the area chair. Good luck.

---

### Meta-Review · Area_Chair1 · 2018-12-13
**Excellent engineering work, but it's hard to see how others can build on it**

**Confidence:** 3
**Recommendation:** Reject

**Metareview:**

This paper describes the development of a large-scale continuous visual speech recognition (lipreading) system, including an audiovisual processing pipeline that is used to extract stabilized videos of lips and corresponding phone sequences from YouTube videos, a deep network architecture trained with CTC loss that maps video sequences to sequences of distributions over phones, and an FST-based decoder that produces word sequences from the phone score sequences. A performance evaluation shows that the proposed system outperforms other models described in the literature, as well as professional lipreaders. A number of ablation experiments compare the performance of the proposed architecture to the previously proposed LipNet and "Watch, Attend, and Spell" architectures, explore the performance differences caused by using phone- or character-based CTC models, and some variations on the proposed architecture. This paper was extremely controversial and received a robust discussion between the authors and reviewers, with the primary point of contention being the suitability of the paper for ICLR. All reviewers agree that the quality of the work in the paper is excellent and that the reported results are impressive, but there was strong disagreement on whether or not this was sufficient for an ICLR paper. One reviewer thought so, while the other two reviewers argued that this is insufficient, and that to appear in ICLR the paper either (1) should have focused more on the preparation of the dataset, included public release of the data so other researchers could build on the work, and put forth the V2P model as a (very) strong baseline for the task; or (2) done a more in-depth exploration of the representation learning aspects of the work by comparing phoneme and viseme units and providing more (admittedly costly) ablation experiments to shed more light on what aspects of the V2P architecture lead to the reported improvements in performance. The AC finds the arguments of the two negative reviewers to be persuasive. It is quite clear at this point that many supervised classification tasks (even structured classification tasks like lipreading) can be effectively tackled by a combination of a sufficiently flexible learning architecture and collection of a massive, annotated dataset, and the modeling techniques used in this paper are not new, per se, even if their application to lipreading is. Moreover, if the dataset is not publicly available, it is impossible for anyone else to build on this work. The paper, as it currently stands, would be appropriate in a more applications-oriented venue.